

# Adaptive control for uncrewed aerial vehicles based on communication information optimization in complex environments

Zirong Wang[1], Zhengyu Han[1] and Shahzadi Tayyaba[2]

[1] Equipment Management and Unmanned Aerial Vehicle Engineering School, Air Force Engineering University, Xi'an, Shaanxi, China
[2] Division of Science and Technology, University of Education, Township Campus, University of Education, Lahore, Pakistan

## ABSTRACT

The utilization of drone technology thrives in diverse domains, including aviation, military operations, and logistics. The pervasive adoption of this technology aims to enhance efficiency while mitigating hazards and expenditures. In complex contexts, the governing parameters of uncrewed aerial vehicles (UAV) require real-time adjustments for flight safety and efficacy. To improve the attitude estimation accuracy, this article introduces a ATT-Bi-LSTM framework for optimizing UAVs through adaptive parameter control, which integrates the state information gleaned from communication signals. The ATT-Bi-LSTM achieves data feature extraction by means of a two-layer Bidirectional Long Short-Term Memory (BI-LSTM) at its inception to enhance the feature. Subsequently, it harnesses the attention mechanism to amplify the LSTM network's output, thereby enabling the optimal control of UAV positioning. During the empirical phase, we employ optical system data for the comparative validation of the model. The outcomes underscore the commendable performance of the proposed framework in this study, particularly with regard to the three pivotal position indicators: yaw, pitch, and roll. In the comparison of indicators such as RMSR and MAE, the proposed model has the lowest error, which provides algorithm support and important reference for future UAV optimization control research.

# INTRODUCTION

The rapid evolution and ubiquitous integration of drone technology stand as conspicuous hallmarks within the modern technological landscape. In tandem with the relentless march of progress, drones have transcended their origins in the military sphere and now find extensive application in civilian and commercial domains. Presently, drone technology exerts a profound influence across diverse sectors, ranging from agriculture, aerial photography, environmental surveillance, and rescue missions to logistics and scientific investigation. Drones furnish an economically prudent and efficacious means to execute tasks, thereby facilitating the profligate consumption of human resources

Corresponding author
Zirong Wang,
Bt-7274555485@outlook.com

and time (*Fan, Li & Zhang, 2020*). Furthermore, these aerial marvels demonstrate their exceptional utility by venturing into perilous or otherwise inaccessible regions, where they proficiently undertake a plethora of tasks, encompassing search and rescue missions, natural disaster monitoring, and epidemic detection. In a complementary vein, drone technology assumes a pivotal role in the agricultural arena, elevating crop management efficiency through the judicious employment of intelligent sensors and cutting-edge image processing technology. Consequently, farmers stand better equipped to administer their arable lands and augment crop yields (*Shi et al., 2022*). Hence, the attainment of streamlined and precision-centric drone control emerges as an indispensable cornerstone in the expansion of the drone industry.

The role of communication data in drone control is pivotal. Effective surveillance of communication information during drone operations holds paramount significance in the evaluation of drone flight stability. The advancement of real-time communication technology has, in tandem, propelled the swift evolution of drones. Deep learning techniques are harnessed for the implementation of visual simultaneous localization and mapping (SLAM), endowing drones with the capability to construct real-time environmental maps and ascertain their positions, thereby facilitating more judicious flight path planning. Moreover, deep reinforcement learning methods assume a pivotal role in autonomous control, enabling drones to deftly adapt to dynamic environmental fluctuations and navigate the vagaries of uncertain flight missions (*Zeng & Zhang, 2017*).

Consequently, they become integral in contexts like search and rescue operations, security monitoring, and military missions. Within this sphere, a succession of improved YOLO methods and novel network architectures have burgeoned, ushering in a burgeoning wave of research from the UAV perspective. The reliability and efficiency of communication data are non-negotiable for the success of drone control. The expeditious progress of deep learning has wrought a profound transformation in the realm of drone pose control (*Wu, Zeng & Zhang, 2018*). Neural network controllers stand to augment flight performance, particularly in the face of inclement weather conditions, through the simulation of flight data. Deep learning applications within the autonomous driving system bolster UAV flight stability, enhancing its resilience in the face of risks and unforeseen exigencies. Subsequently, *via* the employment of deep reinforcement learning methods, Drone Reinforcement Learning (DRL) techniques acquire flight strategies through their interactions with the environment. They utilize deep neural networks to articulate value functions and strategies. Established DRL algorithms like Deep Q Network (DQN) and Deep Deterministic Policy Gradient (DDPG) have found application in UAV control endeavors (*Carrio et al., 2017*).

Hence, considering the current landscape of deep learning research, this study delves into the realm of UAV position control and optimization. It proffers a UAV position optimization model predicated on deep learning networks, with the overarching objective of attaining the acme of UAV position control. The precise contributions of this article are delineated as follows:

1. By incorporating communication data during UAV operations, we establish communication strength information and amalgamate it with traditional inertia data,

culminating in the convergence of multiple data sources to enhance the precision of UAV position estimation and optimization.

2. Drawing upon communication attributes and positional characteristics, we constructed the ATT-Bi-LSTM model, adeptly deploying the BI-LSTM network in conjunction with an Attention mechanism to accomplish optimal position estimation in the offline state of UAV positioning.

3. In the process of model validation, through rigorous comparison with data adhering to the gold standard generated under the aegis of optical systems, our model consistently outperforms both traditional control methodologies and position optimization techniques grounded in machine learning. This pronounced superiority of our model augments its utility in conferring both enhanced precision in UAV control and subsequent analytical undertakings.

In the rest of this article, the related work is presented in Section 'Related works'. Section 'Research Method' gives the establishment process of the ATT-Bi-LSTM model. Experiment results and related analysis is provided in Section 'Experiment details', 'Discussion' concludes.

# RELATED WORKS

Based on the current application status and parameter control issues of drones, this article conducted a literature review on relevant literature. This article mainly studies the classic control methods and their improvements for uncrewed aerial vehicles, and focuses on the current control research in the field of deep learning.

## The attitude control for the UAV

In contrast to various other UAV types, quadrotor UAVs boast a host of advantages, notably their compact structure, operational versatility, and a diminished threat to both nearby personnel and equipment. However, they also exhibit distinctive traits such as underactuation, strong coupling, nonlinearity, and susceptibility to external disturbances. As a result, the investigation into control algorithms for quadrotor UAVs has garnered substantial attention from scholars worldwide. Commencing with the classical PID control method, researchers have introduced a spectrum of advanced control algorithms, each aimed at enhancing system stability and robustness. In the initial phases of quadrotor UAV research, the PID controller devised by *Salih & Moghavvemi (2010)* was pivotal in achieving control over positional attitude. An appealing feature of this approach is its freedom from necessitating the development of a dynamic model or precise UAV parameters. To enhance control efficacy, PID is frequently amalgamated and refined in conjunction with advanced control algorithms. *Efe (2011)* innovatively designed a PID controller and harnessed neural networks to mitigate perturbations, resulting in stable control over UAV position and attitude. *Chowdhary, Wu & Cutler (2012)* introduced an adaptive control algorithm with online learning capabilities, which obviates the need for laborious PID parameter adjustments. This approach capitalizes on historical and real-time data to expedite system error convergence without continuous excitation. To ascertain the tracking capabilities of quadrotor UAVs under significant perturbations, *Das, Lewis*

*& Subbarao (2009)* devised a backstepping controller augmented with an integral term. Experimental findings substantiate the controller's capacity to eliminate steady-state errors and bolster overall robustness. Furthermore, *Mohd-Basri, Husain & Danapalasingam (2015)* employed a radial basis function to estimate uncertain perturbations and designed an adaptive backstepping controller capable of robust performance in the presence of unknown disturbances. *Du, Zhu & Wen (2017)* employed the backstepping technique for UAV formation control, resolving the finite-time convergence of error states to the origin through the application of chi-square system theory and Lyapunov stability theory. *Ma, Qin & Salsbury (2014)* conducted a comprehensive analysis and design of the flight controller, leveraging Model Predictive Control (MPC) to suppress active interference in attitude control. *Alexis et al. (2016)* devised robust MPC controllers for two distinct rotor UAV architectures, optimizing the performance metric while integrating model dynamics and input state constraints, thereby minimizing deviations in the presence of severe disturbances. Model Predictive Control, in particular, finds application in UAV trajectory planning. *Sun et al. (2018)* proposed an MPC-based trajectory planning method for effective obstacle avoidance when suspending a payload from a quadcopter UAV. They formulated a cost function considering load swing angle and obstacle-UAV distance to generate an optimal trajectory that satisfies the prescribed requirements.

## Deep learning based UAV optimization research

*Guo et al. (2019)* employed the DQN algorithm for path planning of UAV lift-off platforms, aiming to maximize data transfer rates. However, this algorithm is solely applicable to tasks with discrete action spaces and is plagued by the issue of overvaluing the value function, which can skew the learning of path planning strategies by intelligent agents. In response, *Wang et al. (2019)* harnessed the Double DQN algorithm (*Wang et al., 2019*) to optimize the flight trajectory of UAV platforms, with the objective of maximizing the downlink rate while ensuring coverage of all ground-based users. The Double DQN algorithm mitigates the problem of overestimation inherent in the DQN value function, although it still falls short in its applicability to tasks with continuous action spaces. In contrast, *Liu et al. (2015)* leveraged the DDPG algorithm to successfully implement deep reinforcement learning in continuous action space for path planning tasks.

Furthermore, as deep learning continues to exert a profound influence on research in the field, researchers have increasingly optimized UAV image data for surveillance and target studies. UAV aerial images are predominantly available in the form of visible light and infrared imagery, yet there remains a scarcity of public datasets for visible light, and infrared datasets are even rarer. Notable among the visible light UAV image datasets are VisDrone (*Ullah et al., 2019*) and UAVDT (*Hourani, Kandeepan & Lardner, 2014*). Given the distinctive characteristics of target clustering within UAV images, *Yi, Wang & Meng (2013)* devised a multi-stage cluster detection network, ClusDet, building upon the R-CNN enhancement algorithm. ClusDet leverages region clustering, slice detection, and scale adaptation to enhance the operational speed and the detection rate of small targets within high-resolution UAV imagery. *Duan et al. (2019)* introduced the CenterNet approach, which conceptualizes positioning as a task of center point detection and its

offset. It employs predicted foci for regression to extract the actual position information from the offset parameters of the center regression. This method markedly bolsters the detection rate of small targets, although it does introduce higher resolution output, thereby affecting inference latency.

From the preceding research, a notable gap emerges in the focus on UAV control optimization. While existing studies predominantly emphasize attitude optimization by enhancing classical controllers, there is a distinct lack of attention to optimizing other critical facets of UAV functionality, including path planning and target detection. Deep learning techniques, renowned for their feature extraction capabilities from unknown data, play a pivotal role in addressing this gap. This study strategically addresses the underexplored realm of UAV parameter optimization and attitude control. By leveraging deep learning methodologies, we not only achieve optimal position control but also elevate UAV control to its zenith by introducing additional information. This highlights a crucial area of consideration and underscores the need for a comprehensive exploration of various dimensions in UAV optimization research.

## RESEARCH METHOD

After explaining the current research status, in this chapter, we will explain the methods and provide a detailed introduction to the construction details and related principles of each module, mainly including traditional PID prediction methods, BI-LSTM network methods, and self attention module enhancement.

### PID control

The proportional-integral-derivative (PID) controller represents a prevalent feedback control mechanism extensively applied in the field of control engineering. Its primary objective is to adjust the system's output to closely match a desired reference value. The PID controller comprises three integral components: the proportional (P), integral (I), and differential (D) parts, each with distinct roles in managing the error's magnitude, its accumulation, and the rate of change, respectively (*Houthooft et al., 2017*). These components are computed as demonstrated in Eqs. (1) through (3):

$$P(t) = K_p \cdot e(t) \tag{1}$$

$$I(t) = K_i \cdot \int_0^t e(\tau) d\tau \tag{2}$$

$$D(t) = K_d \cdot \frac{de(t)}{dt} \tag{3}$$

where $P(t)$ is the output of the proportional part. $K_p$ is the proportional gain, and $K_i$ is the integral gain, and $K_d$ is the differential gain, and these three gains are used to regulate the corresponding output section respectively. $e(t)$ is the error at the current

moment, usually defined as the difference between the desired value and the actual value. $e(t) = r(t) - y(t)$, $\int_0^t e(\tau)d\tau$ is the integral of the error, and $\frac{de(t)}{dt}$ denotes the rate of change of the error over time. PID control can be effectively employed to stabilize the attitude of a vehicle, encompassing pitch, roll, and yaw. By continuously assessing the disparity between the vehicle's attitude angle and the desired reference, the PID controller can finely modulate the control of rudder surfaces or motor outputs, ensuring that the vehicle maintains the desired attitude. This process involves the optimal integration of error by considering various types of errors over different time intervals. These preliminary control algorithms prove highly rational and essential for real-time UAV control, establishing a solid foundation for the precise management of UAV flight dynamics.

## Bi-LSTM networks

Long short-term memory (LSTM) stands as a variant of recurrent neural networks (RNNs) explicitly engineered to address the longstanding challenge of managing long-term dependencies in RNNs. LSTM excels particularly in the domain of handling sequential data. Distinguished from conventional RNNs, the LSTM unit primarily optimizes data for long-term control by means of the "forgetting the gate" mechanism, the implementation of which is delineated in Eq. (4) (*Johnson & Moradi, 2005*; *Shi, Wang & Zhao, 2022*):

$$f_t = \sigma\left(W_f \cdot [h_{t-1}, x_t] + b_f\right) \tag{4}$$

where $W_f$ is the weight matrix. $h_{t-1}$ is the hidden state of the previous time step. $x_t$ is the input of the current time step. The hidden state can then be calculated by Eq. (5),

$$h_t = o_t \cdot \tanh(C_t) \tag{5}$$

where Ct is the state of the cell at t. It is related to the computation of the forgetting gate, which can be calculated by Eq. (6):

$$C_t = f_t \cdot C_{t-1} + i_t \cdot \tanh\left(W_c \cdot [h_{t-1}, x_t] + b_c\right) \tag{6}$$

After completing the above calculation of the relevant state quantities, the output of each cell can be obtained:

$$o_t = \sigma\left(W_o \cdot [h_{t-1}, x_t] + b_o\right) \tag{7}$$

The comprehensive overview of the LSTM standalone cell is depicted in Fig. 1, where we also introduce and elucidate the BI-LSTM method. Recognizing the significance of integrating information from different time periods into the position update process during position optimization, this article advocates the utilization of the BI-LSTM method for in-depth analysis.

BI-LSTM represents an extension of LSTM that enhances the model's ability to consider not only historical data but also anticipate future information when handling sequential data. In the standard LSTM, input sequence information is processed from left to right, while BI-LSTM processes both left-to-right and right-to-left data. This approach equips the model to grasp the entire sequence comprehensively, enabling it to assimilate not only

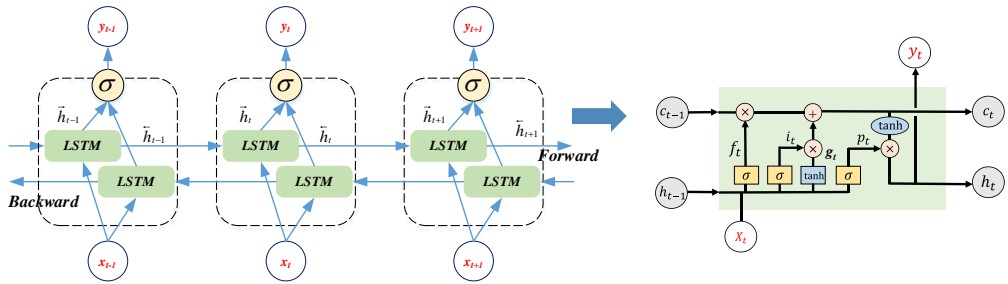

**Figure 1  The Bi-LSTM network framework and the LSTM cell.**

past data but also make predictions regarding future information. The computational procedures for BI-LSTM are elucidated in Eqs. (8) through (9):

$$\vec{h}_t = \vec{LSTM}(x_t, h_{t-1}) = \sigma\left(W \cdot [h_{t-1}, x_t] + b\right) \tag{8}$$

$$\overleftarrow{h}_t = \overleftarrow{LSTM}\left(x_t, \overleftarrow{h_{t+1}}\right) = \sigma\left(W \cdot \left[\overleftarrow{h_{t+1}}, x_t\right] + b\right) \tag{9}$$

where $h_t$ is the time step t of the hidden state, and $\vec{h}_t$ is the output of forward LSTM, and $\overleftarrow{h}_t$ is the output of the inverse LSTM,the two merged outputs in the sequence process is shown in Fig. 1 (*Shi, Wang & Zhao, 2022*).

## Attention mechanism based network as well as parameter control

To facilitate a more comprehensive analysis of the correlation within positional data and to achieve optimal positional control, this study enhances the attention mechanism within the framework of the Bi-LSTM network. The attention mechanism is a fundamental concept in deep learning that empowers a neural network model to assign varying attention weights to distinct segments of input when processing sequential data. This refinement enables more effective processing of critical information. Typically, attention mechanism consists of three main components: query (Q), key (K), and value (V) (*Lei et al., 2023*). Q is a vector utilized to pinpoint the location or the information that requires attention, serving as the foundation for calculating the attention weights, typically derived from the preceding layer of the model. K represents the characteristics of the source data, while V corresponds to the value associated with the key. The operational flow of the attention mechanism is elucidated in Fig. 2 (*Lei et al., 2023*).

Attention mechanism is mainly to calculate the attention score, which is shown in Eq. (10):

Attention Scores $= Q * KT$ (10)

where Q is the query vector and K is the transpose of the key vector. After completing the establishment of the attention mechanism, the overall network model established is presented as follows:

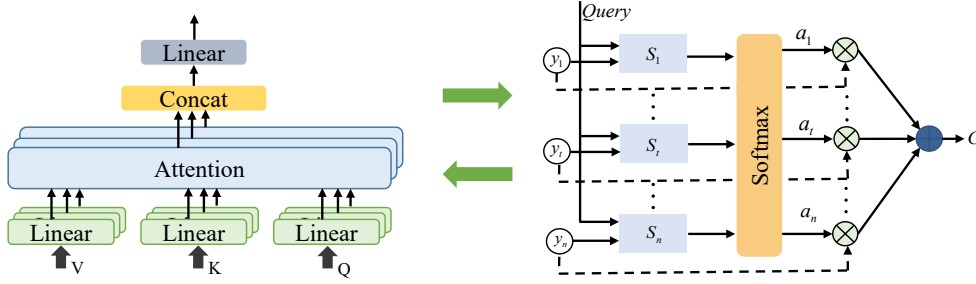

**Figure 2**  The attention mechanism.

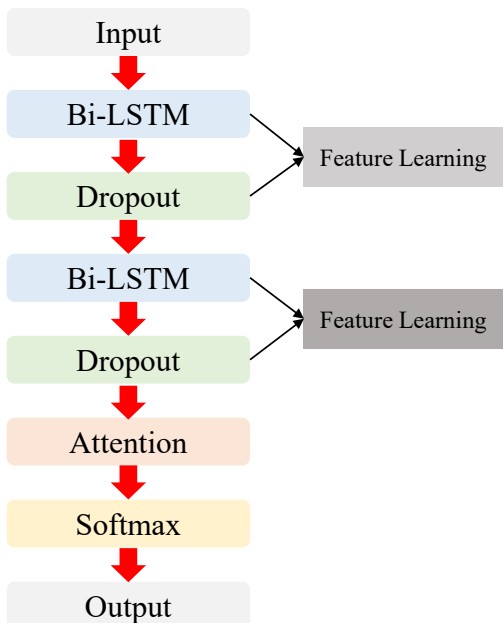

**Figure 3**  The overall structure for the proposed ATT-Bi-LSTM network.

Within the constructed model, the initial input consists of processed UAV position data, which subsequently undergoes processing *via* a two-layer BI-LSTM network in Fig. 3. This phase entails the data's feature extraction through truth-value comparisons. Ultimately, the final step integrates the Attention mechanism to yield the optimized position output.

## EXPERIMENT DETAILS

After building the ATT-Bi-LSTM model, it is necessary to evaluate the effectiveness of the model and complete application analysis. Therefore, we have introduced the selected dataset and provided detailed evaluation indicators and training processes for the model. The above content will be introduced and explained in detail in this chapter.

## Datasets

The wireless signal strength of uncrewed aerial vehicles (UAVs) is contingent upon a multitude of factors, including transmission power (*Wang, Huang & Zhu, 2016*), path loss (*Jang & Lee, 2003*), antenna gain (*Phillips, Sicker & Grunwald, 2012*), and reception sensitivity (*Klein & Degnan, 1974*). Within this context, transmission power determines the power level of the signal dispatched by the UAV, path loss quantifies the reduction in signal power due to distance and environmental influences during signal propagation, antenna gain characterizes the performance of a wireless communication antenna concerning an ideal point source antenna, and reception sensitivity denotes the minimum signal power level that the receiving device can detect and decode.

This article computes the signal strength by taking these four factors into account, as demonstrated in Eq. (11) (*Liu et al., 2015*).

$$\text{Received Power} = \text{Transmit Power-Path Loss} + \text{Antenna Gain-Free Space Loss.} \tag{11}$$

In this equation Path Loss denotes the path loss, Antenna Gain denotes the antenna gain, and Free Space Loss can be calculated by the free space propagation equation:

$$FSL(dB) = 20 \cdot \lg(\frac{4\pi d}{\lambda}) \tag{12}$$

where FSL denotes free space loss, d is distance and λ is wavelength, this equation considers the relationship between wavelength and distance of the propagating signal.

In the current research on UAV attitudes, the data primarily originates from video image data captured during target monitoring, as exemplified by datasets such as the UAV123 Dataset (*Zhang et al., 2019*) and VisDrone Dataset (*Mueller, Smith & Ghanem, 2016*). Nonetheless, these datasets may not fully suffice to meet the study's requirements. Additionally, when examining communication relationships, publicly available datasets may also fall short of the requirements. To address this, this study draws inspiration from *Han, Hu & Zhou (2019)*, which employed an optical verification system to establish a relevant data model. The data collection primarily centers on three types of attitude information of the UAV, namely yaw, pitch, and roll, while also recording communication strength data throughout the entire course of UAV motion. This data serves as the foundation for subsequent model training. In the initial phase of test validation, the PID method, as discussed in Section 'PID control', is employed to realize UAV control and collect pertinent data. Subsequently, in offline simulations, neural network methods are employed to validate attitude and perform data analysis.

## Experiment details

To evaluate the proposed model, the index for the method comparison in this article including the RMSE, MAE, MdAE, MdAPE, which is shown as follows (*Han, Hu & Zhou, 2019*).

$$RMSE = \sqrt{\frac{1}{n}\sum_{i=1}^{n}(\text{Actual}_i - \text{Predicted}_i)^2} \tag{13}$$

$$MAE = \frac{1}{n}\sum_{i=1}^{n}|Actual_i - Predicted_i| \tag{14}$$

$$MdAE = median\left(|Actual_i - Predicted_i|\right). \tag{15}$$

In the above equations, median denotes the median, Actuali denotes the first i observation, $P_i$ denotes the first observed value, In addition to these three indexes, we also evaluated the model using the Median Absolute Percentage Error (MdAPE), which is calculated as follows.

$$MdAPE = median\left(\frac{|Actual_i - Predicted_i|}{Actual_i} \times 100\right). \tag{16}$$

Based on the established evaluation indexes and combined with the overall network model established in the 'Research method' section for model training and evaluation, the overall algorithmic is as follows:

---

**Algorithm 1: Training process of ATT-Bi-LSTM for parameters optimization**

**Input:** Attitude, position and communication information for the UAV
**Initialization.**
Define the ATT-Bi-LSTM.
Define the training information including: initial parameters, optimizer and max training epochs.
**Feature extraction.**
Using the original data of the Attitude, position and communication information.
**Model training:** Epochs initialization.
**while** epoch<set preset epoch  **do**
Sample data from Input.
Feed data to ATT-Bi-LSTM network.
Model updates.

**End**
**Parameters Fine tuning**
**while** epoch<*preset epoch* **do**
Feed validation data to ATT-Bi-LSTM network.
Loss and gradients calculation.
**Compute** RMSE, MAE, MdAE, MdAPE
**Save** the optimal model

**end**
**Output:** Trained **ATT-Bi-LSTM** network

---

## The method comparison

Drawing from the dataset and related data that have been established, this article conducted practical tests. During the model validation process, we also engaged in a comparative

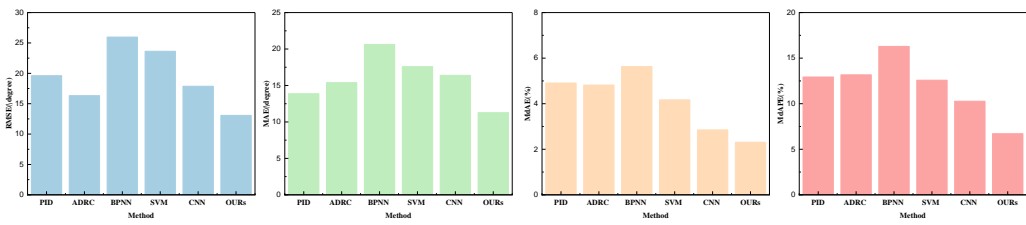

**Figure 4** Performance comparison with other parameters optimization methods on yaw.

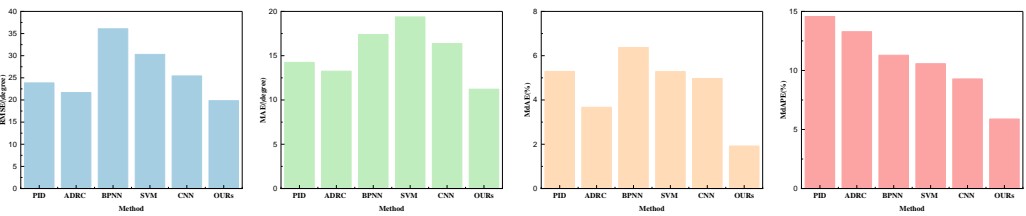

**Figure 5** Performance comparison with other parameters optimization methods on pitch.

analysis of various methods, using the established dataset for this purpose. The comparison encompassed several commonly used methods in the realm of UAV posture study, including PID, ADRC (*Willmott & Matsuura, 2005*), as well as classic machine learning methods like BPNN, SVM, and LSTM, along with more established deep learning methods such as CNN, to assess their performance and suitability. BPNN is an artificial neural network based on backpropagation algorithm, mainly used for regression tasks. Its characteristics include strong nonlinear modeling ability, suitable for fitting complex relationships, but requiring a large amount of data for training and easy overfitting. The application of SVM in regression is based on the support vector regression (SVR) model, which constructs an optimal hyperplane to fit the data. Its characteristics include effectively handling complex relationships in high-dimensional spaces, having good robustness against outliers, but may face significant computational challenges when dealing with large-scale data.

## Experiment results and analysis

In this article, our primary focus centers on the assessment of three key indicators: yaw, pitch, and roll. The outcomes of our analysis are visually presented in Figs. 4–6 and are detailed in Tables 1–3.

Upon evaluating the yaw indicators, it is evident that the proposed method excels across all four indicators. Particularly noteworthy is its remarkable performance in terms of root mean square error (RMSE), which surpasses the performance of traditional machine learning methods and individual CNN feature extraction methods. This underscores the pivotal role played by the double-layer BI-LSTM in both feature extraction and model optimization. Furthermore, the favorable results for MdAEr and MdAPE underscore the

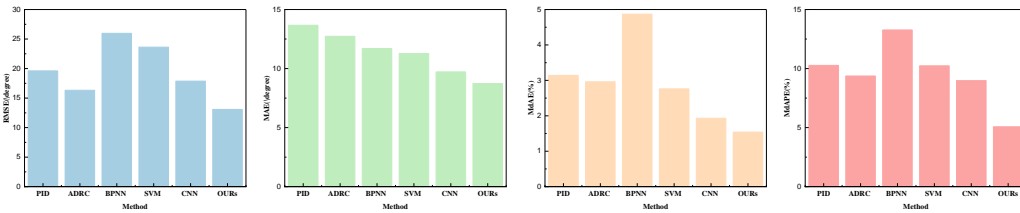

**Figure 6  Performance comparison with other parameters optimization methods on roll.**

**Table 1  Performance comparison with other parameters optimization methods on yaw.**

| Method | RMSE | MAE | MdAE (%) | MdAPE (%) |
|---|---|---|---|---|
| PID | 27.18 | 13.87 | 4.91 | 12.93 |
| ADRC | 22.34 | 15.39 | 4.82 | 13.17 |
| BPNN | 38.79 | 20.63 | 5.63 | 16.29 |
| SVM | 37.54 | 17.58 | 4.17 | 12.58 |
| CNN | 25.67 | 16.39 | 2.85 | 10.27 |
| OURs | 21.14 | 11.27 | 2.31 | 6.71 |

**Table 2  Performance comparison with other parameters optimization methods on pitch.**

| Method | RMSE | MAE | MdAE (%) | MdAPE (%) |
|---|---|---|---|---|
| PID | 23.86 | 14.24 | 5.29 | 14.57 |
| ADRC | 21.67 | 13.25 | 3.67 | 13.28 |
| BPNN | 36.07 | 17.39 | 6.37 | 11.29 |
| SVM | 30.29 | 19.38 | 5.28 | 10.57 |
| CNN | 25.43 | 16.37 | 4.97 | 9.28 |
| OURs | 19.84 | 11.22 | 1.92 | 5.89 |

comprehensive performance advantage offered by this method. Yaw, being a challenging and error-prone aspect of inertial navigation and UAV control, has yielded highly satisfactory control outcomes.

As those three indicators are very important for the control estimation, we made a fine analysis about that. In the comparison of the data pertaining to pitch and roll, as depicted in Figs. 5 and 6, it is discernible that their indicators are notably lower than those for yaw. All indicators attest to robust prediction performance. A thorough analysis of the three indicators—yaw, pitch, and roll—reveals that the employed method exhibits superior pose optimization capabilities, with lower errors compared to actual values collected by optical systems. This substantiates the method's efficacy in optimizing pose. Among these indicators, it is apparent that yaw exhibits the largest error, primarily due to the sensor's susceptibility to strong geomagnetic interference and substantial signal fluctuations, necessitating more comprehensive error correction. In the performance analysis of the aforementioned methods, it becomes evident that the PID method yields

Table 3   Performance comparison with other parameters optimization methods on roll.

| Method | RMSE | MAE | MdAE (%) | MdAPE (%) |
|---|---|---|---|---|
| PID | 19.61 | 13.67 | 3.14 | 10.27 |
| ADRC | 16.32 | 12.73 | 2.96 | 9.37 |
| BPNN | 25.97 | 11.69 | 4.87 | 13.27 |
| SVM | 23.63 | 11.27 | 2.76 | 10.23 |
| CNN | 17.88 | 9.72 | 1.93 | 8.98 |
| OURs | 13.08 | 8.73 | 1.54 | 5.07 |

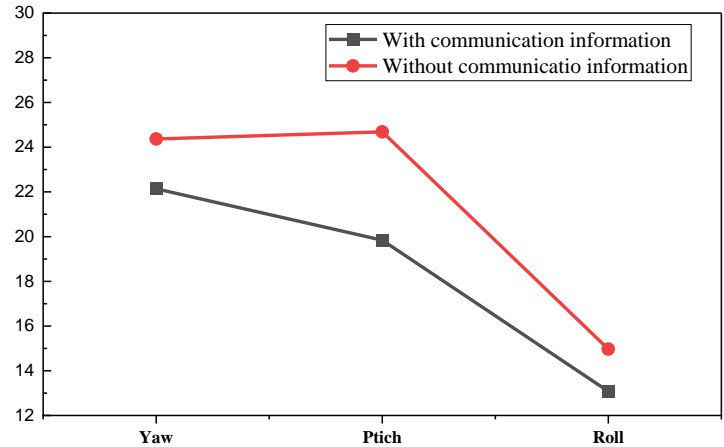

Figure 7   The communication.

moderate performance across all three indicators, indicating its suitability for preliminary control during the early stages of this experiment.

Furthermore, this article incorporates the FSL information introduced at the outset of this chapter into the pose optimization process. By introducing communication information, the model's phenotype can be somewhat enhanced. To better validate the effectiveness of this additional information, the article conducts optimization function tests on the model with and without this information, with results displayed in Fig. 7.

As Fig. 7 illustrates, the overall estimation value in this study exhibits a more pronounced improvement after the addition of communication information. Consequently, in future UAV position control, introducing more communication information will enhance the effectiveness of position optimization.

Regarding the deep model, the data input size exerts a discernible impact on its performance. In this article, subsequent to conducting comparisons of multiple methods, we also executed batch size comparison experiments for the proposed method across various datasets. The results are presented in Fig. 8.

In our experimentation involving diverse batch sizes, a comprehensive analysis was conducted across a spectrum of batch sizes, including 2, 4, 8, and up to 64, with corresponding boxplots constructed to provide a visual representation of the results. The

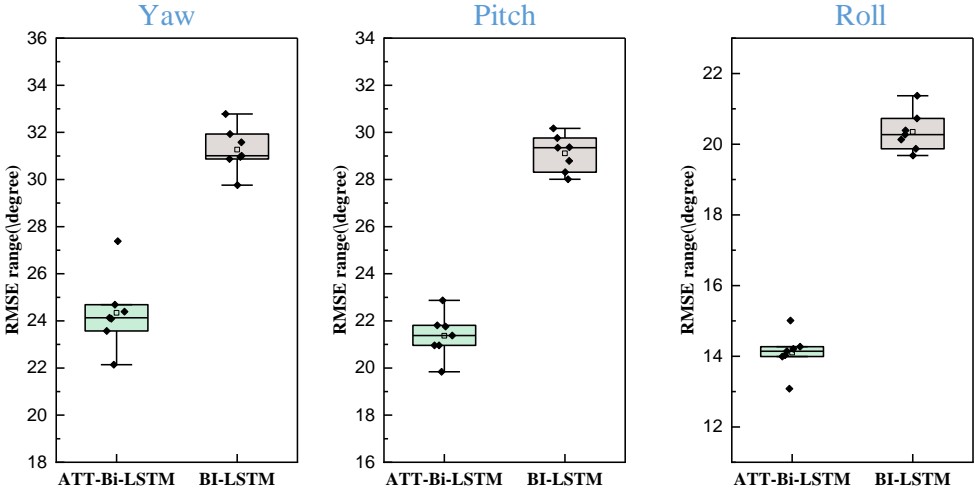

**Figure 8** **The results for the attitude estimation when model with and without attention mechanism.**

findings, as depicted in Fig. 8, reveal a remarkable consistency in the model's performance across the range of batch sizes, showcasing minimal deviation. This robust behavior under varying batch sizes is a noteworthy observation, highlighting the model's stability and adaptability to different input scales. Furthermore, the ATT-Bi-LSTM model consistently outperforms alternative models, demonstrating its superior optimization performance across the entire batch size spectrum. The model's ability to maintain high-quality results across different batch sizes underscores its reliability and efficacy, reinforcing its suitability for diverse scenarios and datasets.

# DISCUSSION

This article addresses the control challenges of uncrewed aerial vehicles in complex environments and introduces an offline BI-LSTM network with an attention mechanism, achieving UAV optimization control and attitude optimization by fusing communication information. In the model verification and comparison process, we utilized the classic PID algorithm as the initial benchmark and subsequently refined and optimized pose control using improved PID and BPNN methods (*Zhou et al., 2020*). The study also conducted a detailed analysis and comparison with the more traditional ADRC method (*Willmott & Matsuura, 2005*). The ADRC method, which incorporates internal and external loop control and employs nonlinear states, offers certain advantages in optimal control. However, this method is hampered by its numerous parameters and challenges in tuning. In real-time applications, ADRC is advantageous for real-time attitude control due to its rapid response and theoretical foundation in pure mathematical methods. The proposed method, along with other machine learning techniques, significantly reduced RMSE and other errors during subsequent pose optimization, indicating the potential for deep learning methods like BPNN, CNN, and LSTM in the realm of uncrewed aerial vehicles. As edge computing

and micro-processing capabilities continue to improve, a broader array of methods can be applied to advance UAV research, yielding enhanced control outcomes.

Future research into UAV optimization using deep learning methods will introduce DRL (*Chen, Liu & Xiong, 2019*) to facilitate the learning of flight strategies through interactions with the environment. UAVs can refine their decision-making processes through experiential learning. For instance, flight strategy optimization can be realized using techniques such as deep Q-network, deep deterministic policy gradient (DDPG), and proximal policy optimization (PPO). Furthermore, the integration of classic object detection methods like YOLO and Faster R-CNN expands their utility. Moreover, as chips become more power-efficient and higher-performing, the reception of satellite information allows for more precise data analysis. These evolving deep learning methods are continually advancing the field of UAV control and navigation, bestowing UAVs with heightened levels of autonomy and intelligence, enabling them to execute more intricate and diverse tasks across a multitude of application domains. Deep learning methods play an important role in the future development of uncrewed aerial vehicles. Firstly, deep learning technology can be used for the perception and decision-making of drones, enhancing their environmental perception capabilities. Through deep learning algorithms, drones can achieve more accurate target detection, recognition, and tracking, enhancing their autonomous navigation capabilities in complex environments. Secondly, deep learning plays a crucial role in the autonomous flight and path planning of drones, enabling them to complete various tasks more flexibly and efficiently, such as search and rescue, monitoring, and cruising. In addition, deep learning can also be used to enhance the adaptability and learning ability of drones, enabling them to dynamically adjust strategies according to different environments and tasks. Overall, the application of deep learning methods will drive the development of drone technology, making it more intelligent, flexible, and adaptable in the future.

Real-time adjustments of UAV parameters are crucial for ensuring flight safety and optimizing efficacy in diverse scenarios. Dynamic environmental conditions, such as sudden weather changes, demand real-time parameter modifications to maintain stability and safe navigation. In complex terrains or urban landscapes, the ability to adapt UAV parameters in real-time is essential for avoiding obstacles and ensuring successful navigation. Surveillance and reconnaissance missions benefit from dynamic parameter adjustments when tracking moving targets, while emergency situations and collaborative UAV operations require rapid tuning for effective response and coordination. Overall, real-time parameter adjustments play a pivotal role in enhancing the adaptability and performance of UAVs across a range of scenarios.

## CONCLUSION

This study presents an optimization model for UAV position control that leverages ATT-Bi-LSTM for accurate attitude estimation, achieving optimal control and attitude estimation during UAV motion. The model's estimation results undergo testing against optical system position errors, and a comparative analysis is carried out using metrics such as RMSE and MAE to meet practical requirements. The experimental findings highlight the significant impact of the proposed network, surpassing the performance of traditional control and machine learning-based position optimization methods. Notably, the incorporation of communication information substantially improves the model's position estimation accuracy, resulting in an impressive average 15% reduction in RMSE across yaw, pitch, and roll indices. This notable enhancement enhances the efficiency and accuracy of the model's estimation, emphasizing its crucial role as a valuable reference and theoretical foundation for future research in UAV control and position estimation.

Although this article has achieved good results in the current pose optimization problem, this type of problem is optimized based on the obtained data. In tasks with real-time requirements, we need to further improve their methods. In future work, we anticipate expanding the diversity of data processing in the current model by incorporating more UAV self-contained information, such as barometric pressure and image data, to enhance UAV control accuracy and broaden its application scope. Although the ATT-Bi-LSTM model utilized in this study exhibits robust estimation and optimization capabilities, further exploration is required to streamline the model's complexity. Furthermore, as UAVs may vary in the number and type of sensors they carry, adapting optimization approaches to the specific information they carry remains a focal point for future research.

### Funding
The authors received no funding for this work.

### Competing Interests
The authors declare there are no competing interests.

### Author Contributions
- Zirong Wang conceived and designed the experiments, performed the experiments, performed the computation work, prepared figures and/or tables, authored or reviewed drafts of the article, and approved the final draft.
- Zhengyu Han conceived and designed the experiments, analyzed the data, performed the computation work, prepared figures and/or tables, authored or reviewed drafts of the article, and approved the final draft.
- Shahzadi Tayyaba conceived and designed the experiments, performed the experiments, analyzed the data, prepared figures and/or tables, and approved the final draft.

## Data Availability

The code is available in the Supplementary File.

The data is available at Zenodo: Antreas Palamas, & Panayiotis Kolios. (2023). Drone onboard multi-modal sensor dataset (Version 1) [Data set]. Zenodo. https://doi.org/10.5281/zenodo.7643456.

## Supplemental Information

Supplemental information for this article can be found online at http://dx.doi.org/10.7717/peerj-cs.1920#supplemental-information.

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
