# Peer review of "Adaptive control for uncrewed aerial vehicles based on communication information optimization in complex environments"

_PeerJ Computer Science, doi:10.7717/peerj-cs.1920_

## Round 0.1 · original submission · Major Revisions

Dear authors,

Thank you for submitting your article. The reviewers’ comments are now available. Your article has not been recommended for publication in its current form. However, we encourage you to address the reviewers' concerns and criticisms; particularly regarding methods, experimental design and validity, and resubmit your article once you have updated it accordingly.

Best wishes,

**Language Note:** The review process has identified that the English language must be improved. PeerJ can provide language editing services - please contact us at [email protected] for pricing (be sure to provide your manuscript number and title). Alternatively, you should make your own arrangements to improve the language quality and provide details in your response letter. – PeerJ Staff

Reviewer 1 ·

Basic reporting

Having thoroughly examined your paper entitled " Adaptive Control for Unmanned Aerial Vehicles Based on Communication Information Optimization in Complex Environments," I commend your efforts in advancing the field. However, I would like to offer some constructive suggestions:

Begin by clearly defining ATT-Bi-LSTM to ensure readers grasp the methodology right from the start.

Smooth Transition Between Sections: Ensure a smoother transition between sections, especially between the introduction and the methodology, to maintain a coherent flow of ideas.

Experimental design

Highlight Practical Implications: Discuss the practical implications of the proposed framework for industries utilizing UAV technology, going beyond the academic context.

Include Limitations Section: Introduce a section that discusses the limitations of your proposed method, acknowledging potential challenges and areas for future improvement.

Address Empirical Data Limitations: Clearly state any limitations or constraints related to the use of optical system data for empirical validation.

Validity of the findings

Consider incorporating visual aids, such as diagrams or charts, to illustrate the ATT-Bi-LSTM framework and its components for better reader comprehension.

Strengthen the conclusion by summarizing the key findings and emphasizing the broader impact of your research on the field.

Maintain consistency in the usage of terminology and acronyms throughout the paper for a more polished presentation.

Conduct a final proofreading pass to ensure the clarity and coherence of the entire manuscript, addressing any potential grammatical or typographical errors.

Additional comments

NA

Reviewer 2 ·

Basic reporting

No comment

Experimental design

No comment

Validity of the findings

No comment

Additional comments

I have the following minor concerns that need to be addressed before any possible recommendation.

1. Establish an area for research by highlighting the importance of the topic, and/or making general statements about the topic, and/or presenting an overview of current research on the subject.

2. Identify a research niche by opposing an existing assumption, and/or revealing a gap in existing research, and/or formulating a research question or problem, and/or continuing a disciplinary tradition.

3. Place this research within the research niche by stating the intent of our study, outlining the key characteristics of your research, describing important results, and giving a brief overview of the structure of the paper.


4. Methodology -- were the techniques used to identify, gather, and analyze the data appropriate to addressing the research problem? Was the sample size appropriate? Were the results effectively interpreted and reported?

5. Although the Results section must provide a detailed description of the data collected, there needs to be a critical synthesis and comparison of the findings in the analysis of the results.

6. Justify Real-Time Adjustments: Elaborate on the specific scenarios where real-time adjustments of UAV parameters become critical for flight safety and efficacy.

7. Include a more extensive comparative analysis with existing methods to showcase the uniqueness and superiority of ATT-Bi-LSTM.

8. Clearly articulate how the attention mechanism amplifies the LSTM network's output, contributing to optimal UAV positioning

---

## Round 0.2 · accepted · Accept

Dear authors,

Thank you for clearly addressing all the reviewers' comments. I confirm that the quality of your paper is improved. The paper is now ready for publication in light of this revision.

Best wishes,

Reviewer 1 ·

Basic reporting

All the concerns have been addressed. The paper can be accepted in its current state.

Experimental design

NA

Validity of the findings

NA

Additional comments

NA

Reviewer 2 ·

Basic reporting

no comment

Experimental design

no comment

Validity of the findings

no comment

Additional comments

The authors have addressed all the comments and the editorial quality has been improved. Therefore, I recommend publishing the paper in the current form.